# Interplay between A-to-I Editing and Splicing of RNA: A Potential Point of Application for Cancer Therapy

**DOI:** 10.3390/ijms23095240

**Published:** 2022-05-08

**Authors:** Anton O. Goncharov, Victoria O. Shender, Ksenia G. Kuznetsova, Anna A. Kliuchnikova, Sergei A. Moshkovskii

**Affiliations:** 1Federal Research and Clinical Center of Physical-Chemical Medicine, 119435 Moscow, Russia; ulteran@gmail.com (A.O.G.); shender_vika@mail.ru (V.O.S.); kuznetsova.ks@gmail.com (K.G.K.); a.kliuchnikova@gmail.com (A.A.K.); 2Faculty of Biomedicine, Pirogov Russian National Research Medical University, 117997 Moscow, Russia; 3Shemyakin-Ovchinnikov Institute of Bioorganic Chemistry of the Russian Academy of Sciences, 117997 Moscow, Russia

**Keywords:** RNA editing, RNA splicing, ADAR, dsRNA, spliceosome inhibitor, type I interferon, cancer therapy, anticancer immunity

## Abstract

Adenosine-to-inosine RNA editing is a system of post-transcriptional modification widely distributed in metazoans which is catalyzed by ADAR enzymes and occurs mostly in double-stranded RNA (dsRNA) before splicing. This type of RNA editing changes the genetic code, as inosine generally pairs with cytosine in contrast to adenosine, and this expectably modulates RNA splicing. We review the interconnections between RNA editing and splicing in the context of human cancer. The editing of transcripts may have various effects on splicing, and resultant alternatively spliced isoforms may be either tumor-suppressive or oncogenic. Dysregulated RNA splicing in cancer often causes the release of excess amounts of dsRNA into cytosol, where specific dsRNA sensors provoke antiviral-like responses, including type I interferon signaling. These responses may arrest cell division, causing apoptosis and, externally, stimulate antitumor immunity. Thus, small-molecule spliceosome inhibitors have been shown to facilitate the antiviral-like signaling and are considered to be potential cancer therapies. In turn, a cytoplasmic isoform of ADAR can deaminate dsRNA in cytosol, thereby decreasing its levels and diminishing antitumor innate immunity. We propose that complete or partial inhibition of ADAR may enhance the proapoptotic and cytotoxic effects of splicing inhibitors and that it may be considered a promising addition to cancer therapies targeting RNA splicing.

## 1. Introduction: Adenosine-to-Inosine RNA Editing by ADAR Enzymes

Before the transcriptional machinery of animal cells provides processing of newly synthesized messenger RNAs by the spliceosome, another important post-transcriptional RNA modification occurs. This is RNA editing by one or more enzymes of the Adenosine Deaminase, RNA-dependent (ADAR) family. Generally, ADARs bind double-stranded secondary structures on RNA and then hydrolytically deaminate adenine bases nearby, which results in the formation of hypoxanthine bases as part of newly formed inosine nucleosides instead of adenosines, respectively [1]. ADAR enzymes bind long, regularly paired RNA duplexes, with multiple adenosines modified as a result; they also bind short, imperfect RNA duplexes, and specific adenosine residues are edited nearby, providing specific editing [2]. Adenosine-to-inosine RNA editing has been described for most types of RNA, including messenger and non-coding RNA species [3].

What are the important changes that are introduced by adenosine conversion to inosine? The fundamental chemical result of this modification is that the exchange of an amino to an oxy moiety changes the ability to form hydrogen bonds [4]. Inosine residues are complementary to cytosine in RNA, not to uracil, mimicking guanosine [5]. Thus, the secondary structure of a transcript is modified after editing; in particular, double-stranded structures are destroyed. In coding parts of mRNA, inosines serving as guanosines in triplets in some cases recode translated protein sequences [6]. A fixed number of amino acid substitutions is generated by this type of mRNA modification, some of which are well characterized and have functional impacts that will be discussed below.

Since the discovery of RNA editing of dsRNA in the Xenopus frog embryo by Bass and Weintraub as far back as 1987 [7], there has been growing interest in the enzymes catalyzing this reaction and in the pathways that the editing is involved in. Soon after the original discovery, it was found that ADAR orthologs are a metazoan innovation and that they are found in many animals, from sponges to mammals [8]. Among the animal kingdom, a number of ADAR-encoding genes vary. Among the well-studied examples in the context of RNA editing, the fruit fly has a single gene for the enzyme, as cephalopods and mammals possess two and three *ADAR* genes, respectively [9].

Structurally, the enzymes contain one or more dsRNA-binding domains, a catalytic deaminase domain and, optionally, a Z-DNA-binding domain with a function unknown until recently (Figure 1) [8]. A major activity of ADARs in organisms in which they have been studied is to deaminate adenosine to inosine in dsRNA sites. This reaction serves two canonical functions of RNA editing. First is the disruption of dsRNA structures. In most known animals, the level of dsRNA in the cytoplasm is subject to innate immunity surveillance by specific sensors that react to excess dsRNA as a potential viral genomic load. Correspondingly, RNA editing has immunosuppressive action and, e.g., works as negative feedback during type I interferon response in mammals, where this induces a production of the cytoplasmic isoform of ADAR1, p150, from an alternative transcription start, which is longer than the constitutive p110 isoform [10]. A second functional impact of RNA editing by ADARs is protein recoding. As inosine pairs more preferably to cytosine in complementary RNA strains, the editing of residues in exonic mRNA parts may cause a fixed number of amino acid substitutions dictated by the genetic code [11], some of them being not conservative and structurally important, such as glutamine to arginine or tyrosine to cysteine. During ontogenesis, recoded proteins are shown to be produced with or even instead of their genomically encoded prototypes and, in a few cases, functional roles have been attributed to these substitutions [6,12,13]. Two above-mentioned functions—immunosuppressive and protein-recoding functions—are, in most cases, specifically performed by different ADAR paralogs, such as ADAR1 and ADAR2 in mammals and cephalopods, respectively, except in the insects, where ADAR1 was lost and functional diversity is reached by alternative splicing and self-editing of the ADAR2 ortholog [14]. In mammals and cephalopods, ADAR1 isoforms contain Z-DNA-binding domains, whereas ADAR2 isoforms and its insect ortholog lack these domains (Figure 1) [9].

In metazoans, significant variation is observed in the protein-recoding function of ADARs. Thus, the exonic sequences of many insects and cephalopods have been shown to be enriched by recoded sites [16,17,18], in contrast to mammals, in which the number of recoded sites is relatively low. It is thought that invertebrates use RNA editing to recode their proteins, mostly in neural and glandular tissues, providing better flexibility during ontogeny accompanied with metamorphosis [19] and for better adaptation to changing ambient temperatures [20,21]. Two complementary theories for the explanation of the evolutionary significance of protein recoding via ADAR editing have been suggested [2]. First, recoding diversifies proteoforms in addition to genomic variation and alternative splicing, which facilitates adaptation during ontogeny and changing circumstances, as mentioned above. Changes in nucleic acid codes that are introduced by RNA editing are even considered as RNA mutations. Indeed, in the context of the RNA world which presumably preceded DNA genomes in early living systems, these RNA mutations represented a driving force in early evolution [22]. A second theory, which does not contradict the first one, states that RNA editing restores functional sequences after deleterious genomic mutations [23]. Evolutionary aspects of A-to-I RNA editing have recently been reviewed in more detail [2,9,24,25].

In correspondence with transcriptome data, a proteome-wide analysis of recoding by ADAR activity has shown that in insects and cephalopods many more proteins have these substitutions than in mammals [16,26,27]. In humans, not more than tens of recoding cases may be identified even in deep proteomic data, the recoded sites being found mostly in the brain [28] and in various tumors [29].

As mentioned above, ADAR enzymes—ADAR1, in particular—are active in most human tissues. Is the RNA modification that they enable essential for survival? In humans, mutations in ADAR1 cause an orphan disease, one of the forms of Aicardi–Goutières syndrome, associated with enhanced interferon responses and inflammation in vital organs [30]. Severe autoimmune responses observed in this syndrome illustrate the vital immunosuppressive function of ADAR1 [31]. Notably, the deadly deficit of ADAR1 in mice may be rescued by knockout of the dsRNA sensor, MDA5 [32]. Similar to ADAR1, mutations of ADAR2 are harmful and, in most cases, fatal during early development [33], despite this isoform being expressed preferentially in the central neural system and in blood vessels in mammals. Very rare cases of patients with this orphan disease, caused by ADAR2 mutations and still unnamed, were recently described [34]. The deficit was accompanied by seizures combined with serious neurological disorder. An essential function of ADAR2 is associated with a single recoded site in the glutamate ionotropic receptor AMPA type subunit 2 (GRIA2), where glutamine is substituted by arginine in position 607 [35]. The resulting substitution decreases conductivity of the corresponding ion channel, which is vital for the developing brain. In ADAR2-deficient mice, a genomic knock-in of a corresponding arginine residue rescued the transgenic animals [36].

The role of A-to-I RNA editing in human cancer has been widely discussed and reviewed elsewhere [37]. Different cancers are characterized by differential expression of ADARs and, correspondingly, various levels of RNA editing [38]. Thus, brain cancers are characterized by low RNA editing [39], whereas hyper- or misediting is reported for some thyroid, head and neck, lung and breast cancers [40]. According to its known immunoregulatory role, the ADAR1 enzyme is generally considered as an oncogenic agent that aids tumors in resisting immunity [38,41]. On the contrary, mRNA editing of the ADAR2 isoform may form neo-antigens through recoding protein sequences and has been attributed a tumor-suppressive ability [42].

Protein recoding via ADAR2 editing of corresponding transcripts was shown to be proapoptotic in esophageal squamous cell carcinoma. Recoding of the IGF-binding protein IGFBP7 downregulated cancer cell proliferation, presumably by inhibition of the Akt pathway [43]. Moreover, the tumor-suppressive action of ADAR2 may be associated, inter alia, with its recently found role in DNA repair. In double-strand DNA breaks, DNA–RNA hybrids are formed which interfere with the DNA-end resection machinery repairing the breaks. ADAR2 has been shown to disrupt the hybrids catalytically interacting locally with BRCA1 and SETX genomic caretakers [44].

Evidence has accumulated that, besides their action via canonical transcript-editing functions, ADARs regulate cancer by editing-independent mechanisms. In glioblastoma, a proto-oncogenic methyltransferase, METTL3, enhanced production of ADAR1 protein via methylation (m6A) of its transcript. Experimental silencing of ADAR1 in this cancer attenuated proliferation, probably through down-regulation of the CDK2 transcript, independently of enzymatic function [45].

Two decades of molecular studies of A-to-I RNA editing have illustrated a crucial role of this post-transcriptional modification in the functioning of human cells and tissues. Further, we will focus more specifically on known interactions of RNA editing and RNA splicing machinery, mostly in the context of their role in cancer. First, we describe an immunoregulatory role of ADAR1 in cancer and how its activity modulates tumor growth. Second, we review a few examples of mechanistic interactions between ADARs and splicing machinery, then describe multiple ways to manage RNA splicing in cancer cells by small molecules. Taken together, these findings make it possible to suggest a hypothesis of simultaneous modulation of both processes which may be beneficial for cancer treatment.

## 2. ADAR1 and Type I Interferon Signaling in Cancer

As mentioned above, an important major function of human ADAR1 is to inactivate intrinsically transcribed dsRNA, which, at excessive levels, can cause innate immune reactions. Transcription of short interspersed nuclear elements, such as Alu elements of the human genome, is a major source of dsRNA in transcripts, as they have potential for intramolecular coupling, with the long dsRNA bands formed resembling some viral genomes. For example, an unprocessed pre-mRNA contains, on average, ten to twenty Alu repeats in its intronic parts [46].

When dsRNAs are transferred to the cytosol, which can occur, for example, as a result of splicing dysregulation, innate immunity sensors, normally responsible for viral nucleic acid recognition, are activated [47]. A major sensor of dsRNA is Melanoma Differentiation-Associated protein 5 (MDA5) which belongs to the group of RIG-I-like receptors [48]. MDA5 recognizes and binds dsRNAs of several hundred nucleotides. After its binding, the activated receptor, via a well-described signaling pathway, induces type I interferon gene expression and secretion of these cytokines [49]. Type I interferons, in turn, act as paracrine and autocrine regulators and, through their receptors, activate the expression of a large pool of specific interferon-stimulated genes (ISGs) [50]. The type I interferon pathway plays a leading role in antiviral response. One group of ISG products arrests the translation of cell and viral proteins, among which protein kinase RNA-activated (PKR) is well described [51]. Eventually, continuous type I interferon signaling arrests cell division and even may lead to apoptosis, which is relevant to cancer cells. Thus, splicing dysregulation in cancer cells, as well as viral attack, such as oncolytic viral treatment, may be beneficial for disease control. At the same time, type I interferons stimulate expression of some genes encoding immunosuppressive proteins as part of a negative feedback loop in response to the main interferon effects [52]. As mentioned above, type I interferon signaling provokes the alternative splicing of ADAR1 transcripts. As a result, a long p150 isoform of this enzyme is produced which contains a cytosol transport signal and, in cytosol, can deactivate dsRNAs as part of the negative feedback response to interferons. Its deaminating enzymatic activity decreases levels of dsRNA in the cytoplasm and, accordingly, immunostimulatory signaling by dsRNA sensors (Figure 2). In addition to RNA editing, the p150 isoform contains a Z-DNA/RNA-binding domain (Zα) which is shown to bind and stabilize RNAs with the Z-RNA conformation in stress granules formed during type I interferon response [53]. The conservation of mRNA in stress granules makes it possible to continue protein translation after its temporary arrest caused by PKR activity [54].

Some cancers specifically need the negative regulation of interferon signaling. Even without immune cell infiltration of the tumor, cancer cells may produce type I interferons [55]. They often demonstrate RNA splicing dysregulation [56] as well as genomic instability [57]. As a result, enormous levels of nucleic acids appear in cytosol [57]. In case of excessive dsRNAs which may reside in mRNA with retained introns, cytosolic ADAR1s can rescue cancer cells from interferon-induced apoptosis [31,58].

Thus, in cancers, the interferon-induced ADAR1 isoform can have pro-cancer effects in two modes. First, diminishing interferon signaling has an antiapoptotic, proliferative action inside the cell. Second, inhibition of paracrine interferon signaling makes the tumor microenvironment immunotolerant to cancer cells. Indeed, it was recently confirmed in patients that a lack of ADAR1 expression can positively correlate with the success of immune checkpoint blockades [59,60]. Moreover, a shorter p110 ADAR1 isoform was recently shown to leave the nucleus upon its phosphorylation under cellular stress conditions [61]. It then rescues some anti-apoptotic gene transcripts from decay, thereby preventing apoptosis of stressed cells [61]. Thus, a blockade of ADAR1 activity is now considered a promising cancer therapy to use along with immunotherapies [59,60]. In addition to the effects of ADAR1 blockade on cancer cells, co-inhibition of this gene and the hnRNPC splicing regulator by CRISPR/Cas9 was recently shown to increase type I interferon response in an acute monocytic leukemia cell line [61]. Apparently, this effect was produced via release of dsRNAs originated from *Alu* repeats into cytoplasm [62].

At the present time, ADAR1 inhibition in model systems is reached by nucleic acid manipulations, such as gene knockouts and knockdowns with the CRISPR/Cas9 system, shRNA or sgRNA [58,60,62,63]. Adenosine analogs, 8-azaadenosine and 8-chloroadenosine, have been suggested as ADAR1 inhibitors [64,65] (Table 1). However, they do not possess enough selectivity to be considered as therapeutics [66]. Further, erythro-9-(2-hydroxy-3-nonyl) adenine hydrochloride, an inhibitor of metabolic adenosine deaminase, an ADA enzyme of different family and origin, was tested for its ability to inhibit ADAR2 and failed to influence the crucial editing of the human GRIA2 glutamate receptor in spite of its ability to inhibit some sites on serotonin receptor (5-HT2CR) mRNA [67]. Recent studies illustrate a need for drug leads which may selectively inhibit ADAR1 to be trialed in the development of cancer therapies.

## 3. Interdependence of A-to-I Editing and Splicing of mRNA

Adenosine-to-inosine conversion, which is catalyzed by ADARs, is able to modify well-described splicing sites on RNAs as they contain functional adenosine and guanosine residues. Obviously, additional 5′-donor and 3′-acceptor sites of introns may be created, as inosine may serve as guanosine there. Further, the AG motif in the acceptor site may be destroyed as well as the adenosine-containing branch point sequence (BPS). Thus, ADAR activity may lead both to the formation of new exons and to exon skipping. Experimental data confirmed examples of splicing modulation by ADAR editing more than decade ago, back in the pre-omics period. Thus, ADAR2 was shown to edit its own transcript by so-called auto-editing, leading to the formation of the new 3′-acceptor splice site at intron 4 of the transcript [68]. Soon after a transcriptome-wide analysis of A-to-I editing became feasible, it was shown that dramatic effects of editing and splicing are not explained only by recoding of 5′ and 3′ cis-elements and BPS [69]. Based on an analysis of different subsets of RNA, from unprocessed early to polyadenylated transcripts, it was demonstrated that ADAR editing generally occurs in the nucleus before splicing, with 95% of editing events preceding polyadenylation [70]. A well-known example of functionally crucial GRIA2 glutamate receptor Q/R editing illustrates this fact. This edited site in the transcript represents a double-strand site formed between a corresponding exonic sequence and a nearby intronic sequence, the latter being essential for editing by ADAR2 [71]. Thus, if the intron excision preceded ADAR2 action, this site would not be edited. 

Knockout murine strains have recently provided models for transcriptome-wide studies of editing effects and splicing. ADAR1 and ADAR2 knockout mice rescued by additional transgenic manipulations, as described above, demonstrated significant perturbation in the splicing landscape [72]. ADAR1 was shown to be responsible for the majority of splicing modulation events, although some of them were provided by ADAR2, for example, an intron 42 retention in the filamin A transcript, a well-described target of ADAR2 editing [73]. Using mutants lacking ADAR1 enzymatic activity, it was found that some of the splicing perturbations were provided independently of the editing process, by physical interactions with splicing machinery [72].

Using knockdowns and other genetic manipulations with human cancer cell lines, Tang et al. were able to elucidate some delicate details of interaction between ADARs and splicing machinery [74]. In an exemplary transcript, CCDC15, both ADAR1 and ADAR2 prevented exon 9 inclusion due to dsRNA binding by both isoforms and to editing activity of ADAR1. The latter edited a specific inhibitory splicing site situated in nearby intron 8. After A-to-I editing, this site was more affine to inhibiting SRSF7 splicing factor [74]. An intronic polypyrimidine (py) tract situated near BPS was shown to form dsRNA bands where ADARs can bind. Thus, in the RELL2 transcript, ADAR2 competed for py tract binding with the U2AF65 splicing factor, which led to exon 3 exclusion and, eventually, to elimination of the resultant transcript by nonsense-mediated decay (NMD) [74]. Exon skipping in the above examples was shown to affect the growth of cancer cells. A proteoform of CCDC15 including exon 9 was found to be oncogenic, and ADARs acted as tumor suppressors in this case, with an opposite role in the regulation of RELL2 proteoforms [74].

As long non-coding RNAs also experience splicing, it was expected that they would be subject to ADAR regulation. Indeed, these effects were found transcriptome-wide in human early embryo development [75].

As mentioned above, ADARs act on transcripts based on their binding to specific conformational structures, mostly dsRNA bands. Despite some evidence that RNA editing precedes splicing in most transcripts [70], situations where alternative splicing affected the conformation of spliced transcripts and thereby modulated editing have been described. These observations related to ADAR2 activity in the mammalian central neural system [76,77]. For example, an osmosensitive cation channel, Tmem63b, had a brain proteoform with skipped exon 4 and Gln-to-Arg recoding via ADAR2 RNA editing in exon 20. These two events were found to be dependent. Retention of exon 4 disrupted a hairpin structure in the exon 20 prone to ADAR2 action [76]. Notably, recoding which leads to an amino acid substitution in the ion channel significantly affects the current passing through the channel and has functional consequences [77].

Above, we have described examples in which ADAR binding modulated the activity of splicing machinery. Recent studies have shown that a major splicing factor, SRSF9, inhibited the editing of sites specific for this isoform, such as brain-specific editing sites in primates [78]. Physical interactions of SRSF9, ribonucleotide reductase regulatory subunit M2 (RRM2) and ADAR2 prevented ADAR2 dimerization, which is necessary for its enzymatic function [78]. Similar inhibitory action of SRSF9 on ADAR2 was found in the editing site of exon 41 in the transcript of the voltage-gated calcium channel CaV1.3 [79].

The data accumulated illustrate many instances of ADAR enzymes modulating RNA splicing, resulting in the skipping and retention of alternatively spliced exons, and, vice versa, protein splicings that are able to change the conformation of transcripts or physically compete with ADAR2 for binding sites, thereby preventing the editing of some sites. All these events of interplay between RNA editing and splicing may cause various biological effects depending on the cells and tissues in which they occur and the functions of alternatively spliced and edited protein isoforms. Thus, the production of isoforms may have both tumorigenic and tumor-suppressive effects, inducing changes in the synaptic conductivity of the brain and other effects not yet described.

## 4. Anticancer Therapeutics Targeting the Spliceosome Machinery

As long as RNA splicing abnormality is shown to be characteristic for many cancer types, the spliceosome, which is a ribonucleoprotein complex responsible for pre-mRNA splicing, is considered an attractive target for new anticancer therapies [80]. More than 20 compounds have been described that are able to inhibit or modulate RNA splicing (Table 1). The first candidate drugs in this class were suggested way back in 1992, whereas the mechanism of their action was not detailed until 2007 [81]. Splicing inhibitors can either block early or late stages of spliceosome assembly or inhibit specific enzymes that catalyze post-transcriptional modifications facilitating RNA splicing [81,82].

The majority of small-molecule splicing inhibitors, such as spliceostatin A, pladienolide B, herboxidiene and their derivatives, block the early stages of spliceosome assembly, more specifically, the assembly of the SF3b complex (Figure 3). The latter is part of the U2 ribonucleoprotein complex (U2 snRNP) and consists of U2 small nuclear RNA and SF3B1-SF3B6 and PHF5A core proteins. The U2 snRNP complex is assembled in proximity to the 3′-splice site of an intron and binds the branch point sequence in an ATP-dependent fashion. Using immune pull-downs and photo-crosslinking, SF3B1 protein was identified as a major target for splicing modulators [83,84,85]. Later, cryogenic electron microscopy confirmed the binding of splicing inhibitors with residues of SF3B1 and PHF5A proteins which are responsible for the formation of a branch point adenosine binding pocket [86,87]. In spite of the fact that these compounds have a common target, the inhibitors modulate alternative splicing of mRNA in various manners and induce different cytotoxic effects in tumor cells [86,88,89,90].

In comparison to cancer cells, the cells originating from normal tissues possess higher resistance to small-molecule splicing inhibitors [91,92]. This observation can be explained by the competition of splicing sites on different RNAs for spliceosome binding. As the compositions of expressed mRNA and produced splicing factors vary between different cells, treatment by the same inhibitor may have different consequences for cell survival. For example, in the presence of the inhibitor, splice sites with weak affinity to splicing factors and less abundant transcripts may lose in the competition for spliceosome binding, which may lead to intron retention. Thus, pladienolide B was shown to support production of pro-apoptotic isoforms of MCL1 and BCL-X in chronic lymphocytic leukemia cells but not in normal blood cells [92]. In addition, a correlation between the potency of splicing modulators and the relative rates of induction for exon skipping and intron retention events was measured. Spliceostatin A and pladienolide analog E7107 were more potent and caused more intron retention events, whereas less potent inhibitors, such as herboxidiene and sudemycin D6, generally induced exon skipping [86]. The precise mechanisms that lead to cancer cell death after splicing inhibition remain unclear. Spliceosome inhibitors alter the profile of intron retention and exon skipping in transcripts. This could, for example, lead to unproductive splicing and subsequent nonsense-mediated decay of DNA repair transcripts (CHEK2) [89] or to the generation of pro-apoptotic protein isoforms (Mcl-1S) [93]. Another possible mechanism could involve a large number of transcripts with retained introns forming an excess of dsRNA in cytoplasm, sensors of dsRNA then activating antiviral signaling and eventually leading to apoptosis of cancer cells [94]. At this point, one could find a connection with the immunosuppressive function of ADAR1, which impairs cellular dsRNA response (see Section 2).

Based on screening of chemical libraries, a new splicing inhibitor, isoginkgetin, was recently identified. Although a physical target of this compound has not yet been found, it is thought to block binding between complex A and complex U4/U5/U6 tri-snRNP during late spliceosome assembly [95].

Some cancer types are especially sensitive to splicing inhibitors and MYC-driven tumors should be mentioned among them. MYC oncogene activation increases transcription rates, requiring a high spliceosomal activity that is vulnerable to inhibitors [80,91]. Synthetic lethality has also been found for some tumors with mutations in splicing factors, such as SF3B1, U2AF1 and SRSF2, in combination with splicing inhibitors [80].

In addition to inhibitors that interact with splicing machinery physically, some compounds acting on other targets can also block or modulate RNA splicing. Among them, histone deacetylase inhibitors (suberoylanilide hydroxamic acid, splitomicin, dihydrocoumarin) and histone acetyltransferase inhibitors (garcinol, AA, BA3) may be mentioned. Dysregulation of spliceosome protein acetylation is thought to inhibit spliceosome assembly [96,97]. Moreover, many spliceosome proteins are regulated by phosphorylation. In this context, CLK and DYRK kinase inhibitors are also considered potential splicing inhibitors [98].

Some antibiotics, such as chlortetracycline, streptomycin and erythromycin, have been reported to inhibit RNA splicing [99]. However, similarly to kinase and acetyltransferase inhibitors, they are less specific than classical spliceosome inhibitors, as described above. Streptomycin and chlortetracycline are affine to pre-mRNA and, at high concentrations, can block early spliceosome assembly. In turn, erythromycin inhibits complex C formation at later stages of spliceosome assembly [99]. Among antibacterial drugs, indisulam, a sulfonamide compound, is considered more promising [100,101]. Indisulam and structurally related sulfonamides, E7820, CQS, and tasisulam, are shown to facilitate the recruitment of splicing factor RBM39 to the DCAF15 E3 ubiquitin ligase, which leads to its polyubiquitination, subsequent proteasomal degradation and, eventually, disturbance of RNA splicing. Tumors of the hematopoietic and lymphoid tissues with high expression levels of DCAF15 are especially sensitive to these sulfonamide compounds [102,103].

Spliceosome inhibitors were successful in preclinical animal trials. One of them, E7107, entered clinical trials for some solid cancers. However, the compound demonstrated excessive toxicity and a lack of anticancer effect, leading to the cancelation of the trials [104,105]. A pladienolide analog, H3B-8800 is currently being trialed for the treatment of myelodysplastic syndromes, acute myeloid leukemia and chronic myelomonocytic leukemia accompanied by mutations in splicing factors [106,107].

**Figure 3 ijms-23-05240-f003:**
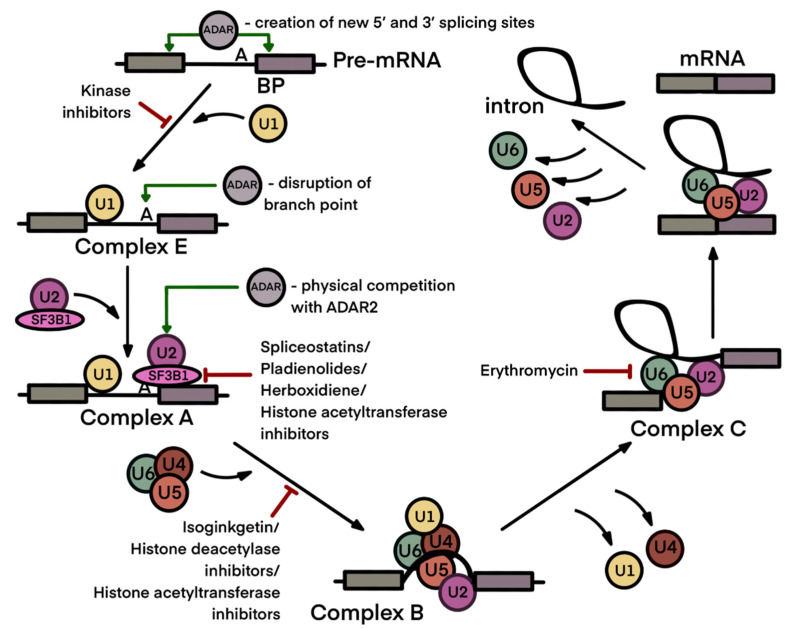
Pre-mRNA splicing process with application points of ADAR enzymes and spliceosome inhibitors. Application points of ADARs and spliceosome inhibitors are designated by green and red arrows, respectively. The pre-mRNA splicing process includes two steps: the recognition of the splicing sites at intron–exon junctions and a branch point sequence (BP), and intron removal and exon end joining. The splice sites are recognized by the U1 small nuclear ribonucleoprotein (snRNP) together with splicing proteins SF1 and U2AF, forming the E complex (early spliceosome). Then, SF1 is replaced by the U2 snRNP to form the A complex (pre-spliceosome). SF3B1 in the U2 snRNP stabilizes the duplex between U2 snRNA and the BP, which is a signal for the recruitment of U4/U6. U5 tri-snRNP enables the formation of the B complex. Then, the spliceosome undergoes a series of intermediate states (Bact, B*, C, C*, P and the intron lariat spliceosome). Both mammalian enzymatically active ADAR isoforms may modify 5′ and 3′ splicing sites and a branch point sequence. ADAR2 may physically interfere with the U2 spliceosome complex (see Section 3). Splicing inhibitors predominantly affect spliceosomal assembly at the early stages (see Section 4). Figure composed based on [108], with amendments.

Splicing inhibitors, especially those which physically interact with a spliceosome at different stages of its assembly are promising for anticancer therapy. The exact mechanisms of their cytotoxic action, as well as the molecular features of sensitive tumors, have yet to be elucidated. In connection with the topic of this review, an important consequence of spliceosome blockade is an excess of dsRNA in the nucleus and cytoplasm, this RNA representing a target for ADAR enzymes. As far as we know, the interaction between splicing inhibitors and ADAR editing has not yet been studied systematically.

## 5. Conclusions and Perspectives

Adenosine-to-inosine RNA editing is an evolutionally ancient type of post-transcriptional modification which is catalyzed by the ADAR family of enzymes and targets double-stranded sites of various RNAs. Deaminating adenosines in long, regularly paired RNA duplexes, the ADAR1 human isoform plays a role in mitigating innate immune response against these duplexes. In contrast, ADAR2 tends to edit shorter and less regular duplexed RNA sites and is responsible for protein sequence recoding via adenosine deamination of codons. 

RNA editing by ADAR enzymes usually precedes RNA splicing in human cells. Working post-transcriptionally with the same substrate, often in overlapping sites, editing and splicing machineries closely interact. In cancer, many examples have been described of the editing of pre-mRNA by one or two active ADAR isoforms leading to the formation of either oncogenic or tumor-suppressive splice isoforms of various proteins. The personalized character of these modulations of splicing makes it literally impossible to target them by any therapy. However, there is a clue with respect to cancer therapy that is more general and which points to a means of overcoming the personalized nature of cancer. This is the immunity that is activated inside and outside the cell with dysregulated splicing via the type I interferon pathway.

Based on the many recent studies cited in this paper, we propose that, in tumors where this enzyme is highly abundant, the enzymatic action of ADAR1 in cytosol contributes to the mitigation of immune responses and the inefficiency of spliceosome-blocking therapies. To conclude the review, we hypothesize that complete or partial inhibition of ADAR1 may enhance the proapoptotic and/or cytotoxic effects of splicing inhibitors. A synergy between two types of action will be reached, as long as the cytosolic dsRNA formed due to splicing inhibition is not inactivated by ADAR1, which will provide a stronger antiviral-like response (Figure 4).

In support of our suggestions, an experimental proof of our hypothesis which proposes a synergy between the inhibition of spliceosomes and ADAR1 was recently provided: the co-inhibition of ADAR1 and hnRNPC splicing factor following gene knockout dramatically increased type I interferon induction [62]. This would be particularly beneficial in the design of new potential cancer immunotherapies.

## Figures and Tables

**Figure 1 ijms-23-05240-f001:**
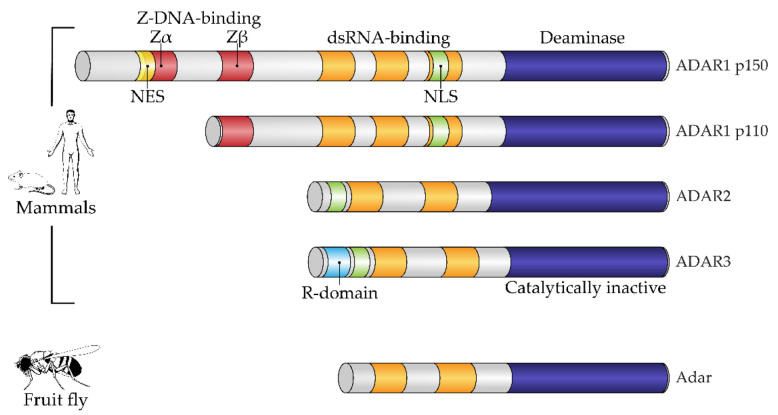
Domain architecture of mammalian and fruit fly ADAR family members. All proteins of the ADAR family share a common domain architecture and include a variable number of dsRNA-binding domains (dsRBDs) and a catalytic deaminase domain. In addition, human ADAR1 isoforms and their orthologs also contain Z-DNA-binding domains. A constitutive isoform, ADAR1 p110, contains the so-called Zβ domain, which adopts a fold similar to Zα domains but is not able to bind a Z-form of nucleic acid. Interferon-inducible ADAR1 p150 has an extended N-terminus containing a Zα domain, which harbors a nuclear export signal (NES). ADAR3, a catalytically inactive member of the family, additionally contains an R-domain. Fruit flies have a single Adar, which is most similar to mammalian ADAR2. NES, nuclear export signal; NLS, nuclear localization signal; R-domain, arginine-rich domain. Figure composed based on [15], with amendments.

**Figure 2 ijms-23-05240-f002:**
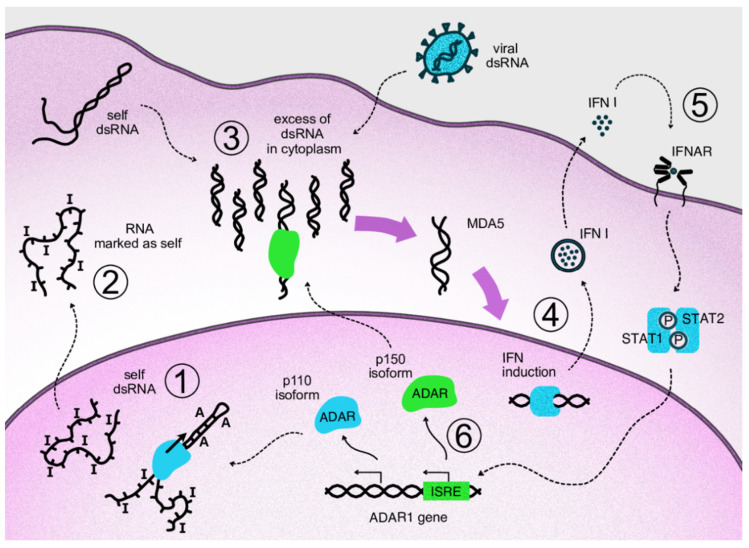
Negative feedback loop of cytosolic ADAR1 isoform acting on type I interferon signaling via inactivation of dsRNA. Repetitive elements of the genome produce transcripts capable of forming double-stranded RNA structures. In the nucleus, these structures undergo ADAR-mediated deamination (1). The cell perceives modified transcripts as self, so their appearance in the cytoplasm does not lead to activation of the dsRNA reactive pathway (2). If levels of dsRNA in cytoplasm are increased as a result of viral attack or overproduction of self dsRNA, e.g., via overexpression of Alu repeats, corresponding sensors are activated, such as MDA5 and RIG-1 (3). Activation of the sensors triggers antiviral innate immunity through transcription factors that initiate type I interferon expression (4). Autocrine and paracrine interferon signaling leads to the transcriptional activation of interferon-stimulated genes via phosphorylated STAT1/STAT2 complexes binding specific genomic ISRE elements (5). In the ADAR1 gene, interferon signaling switches expression to the p150 isoform, which migrates to the cytosol and edits excess dsRNA, thus acting on type I interferon signaling as part of a negative feedback loop (6).

**Figure 4 ijms-23-05240-f004:**
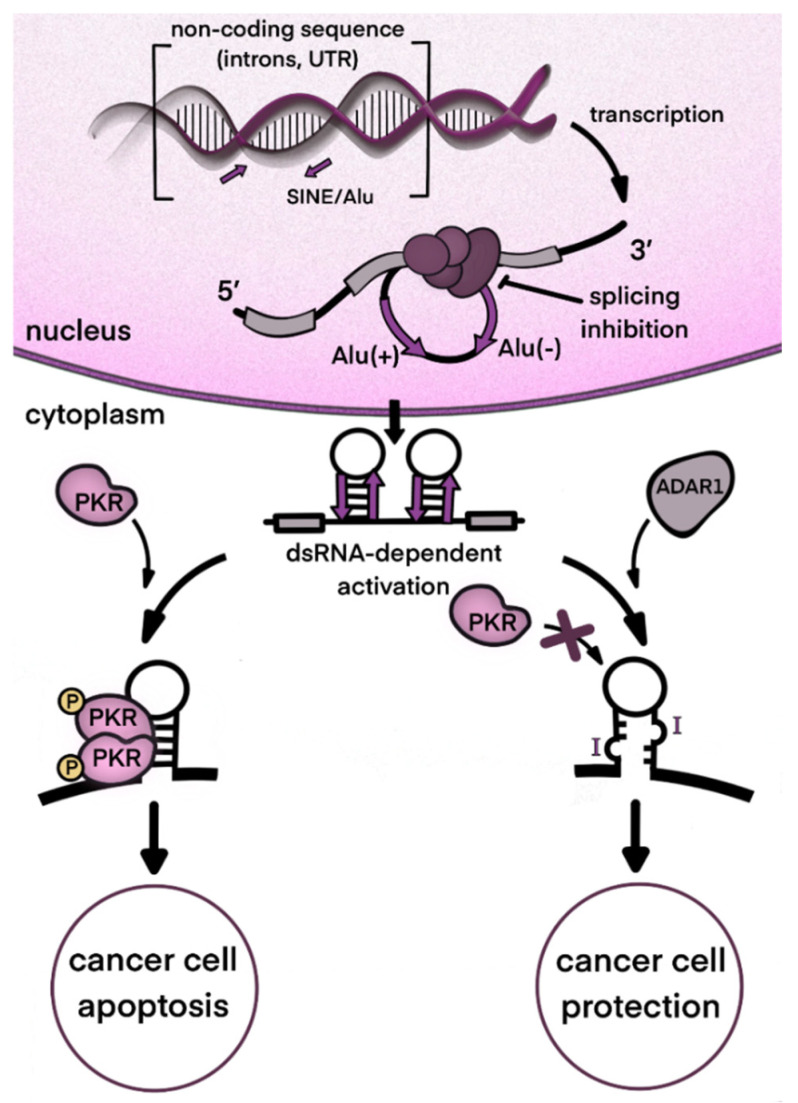
Potential synergy between inhibition of ADAR1 and splicing in cancer therapy. One of the proposed mechanisms of spliceosome-targeted therapies is the accumulation of dsRNA in the cytoplasm. Excess dsRNA activates innate immunity and can lead to PKR-dependent apoptosis. ADAR1 activity destabilizes dsRNA structures and prevents excessive PKR activation, protecting tumor cells and reducing the effectiveness of therapy. ADAR inhibition can significantly enhance the effectiveness of therapy by increasing the amount of dsRNA in the cell and, consequently, enhancing RKR-dependent apoptosis.

**Table 1 ijms-23-05240-t001:** Small-molecule splicing and ADAR editing inhibitors.

Family	Compound	Target/Effect	Clinical Trials
RNA splicing inhibitors			
Spliceostatins	Spliceostatin A	Target SF3B1, block A complex assembly	-
	Sudemycin D6		-
	Meayamycin B		-
Pladienolides	FD-895		-
	Pladienolide B		-
	E7107		+
	H3B-8800		+
Herboxidiene	Herboxidiene		-
	GEX		-
	18-Deoxyherboxidiene		-
Isoginkgetin	Isoginkgetin	Prevents binding of U4/U5/U6 tri-snRNP to the A complex	-
Histone deacetylase (HDAC) inhibitors	Suberoylanilide hydroxamic acid	B complex	-
	Splitomicin		-
	Dihydrocoumarin		-
Histone acetyltransferase inhibitors	Garcinol	A complex	-
	AA	B complex	-
	BA3		-
Kinase inhibitors	Diospyrin	H/E complex	-
	Chlorhexidine		-
	TG003		-
Antibiotics	Chlortetracycline	Early spliceosome assembly	-
	Streptomycin		-
	Erythromycin	C complex	-
Sulfanilamide	Indisulam	Degradation of splicing factor RBM39	+
	E7820		-
	CQS		-
	Tasisulam		+
ADAR RNA-editing inhibitors			
Adenosine analogs	8-Azaadenosine	ADAR1	-
	8-Chloroadenosine		-
Adenine analog	Erythro-9-(2-hydroxy-3-nonyl) adenine hydrochloride (EHNA)	ADA, a metabolic adenosine deaminase; EHNA also inhibited some ADAR2 editing events	-

## Data Availability

Not applicable.

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
