# Peer review of "Interplay between A-to-I Editing and Splicing of RNA: A Potential Point of Application for Cancer Therapy"

_ijms, 2022, doi:10.3390/ijms23095240_

Round 1

Reviewer 1 Report

The review is devoted to the description of RNA editing and splicing, and influence of their balance on cancer. This manuscript develops the theme that in tumors with lots of ADAR1 the spliceosome-blocking therapy is inefficient due to action of ADAR1. Authors propose that simultaneous suppression of ADAR1 and splicing may enhance proapoptotic and cytotoxic effects in tumors. Their suggestion is supported by the recent study of the co-inhibition of ADAR1 and hnRNPC which induced type I interferon induction. The paper describes in detail properties and features of adenosine-to-inosine RNA editing by ADAR. Further, authors focus at the negative feedback loop of cytosolic ADAR1 isoform acting on type I interferon signaling via inactivation of dsRNA. The next section is devoted to the connection of two mechanisms – RNA editing and mRNA splicing. Then the big section describes approaches of cancer therapy based on spliceosome inactivation. In the final part of the work, the possible importance for cancer therapy of the cooperative suppression of ADAR1 and splicing is described. As a result, the authors convincingly demonstrated that adenosine-to-inosine RNA editing by ADAR and mRNA splicing are bound and could be used as targets for cancer therapy. It should be noted that this review is extremely relevant and summarizes the latest data on RNA editing and mRNA splicing. The review is beautifully illustrated, written clearly and concisely. I believe that it will undoubtedly be of interest to the widest range of readers, as it illuminates one of the fundamental processes taking place in cells.

Author Response

We thank the Reviewer for the comprehensive and positive review of our manuscript.

Reviewer 2 Report

This is an interesting topic (RNA editing, RNA splicing and cancer),
which would best suit Cancers Journal at MDPI. The link to molecular
sciences is rather vague. This is probably due to lack of clarity in
text. For instance, abstract reads relatively well up to line 20. Then
we rather completely lost the topic.
“In turn, a cytoplasmic isoform of ADAR1 can deactivate dsRNA in
cytosol, thereby interfering with type I interferon response and
providing resistance to splicing inhibitors”. What does this text
mean? Same remark for “we conclude that complete or partial inhibition
of ADAR1 may enhance proapoptotic and cytotoxic effects of splicing
inhibitors and may be considered as a promising addition to cancer
therapies targeting RNA splicing”. This probably addresses a qualified
audience at an oncology journal. Besides, a review cannot be
conclusive. It can only be descriptive and discuss some new
perspectives. The review for molecular sciences may discuss about ADAR
and other DNA/RNA editing enzymes, but even ADAR1 is not properly
presented here. Editing enzymes are not really neglected by textbooks.
This is rather emerging field due to the accumulation of data on
ADAR1/ADAR2/ADAR3/Apobec structure and their mode of action or target,
not only in human but also in insect, squid and bacterial organisms. Reviewing literature is rather incomplete here. There are more simple things
than “ADARs bind double stranded secondary structures on RNA and then
they hydrolytically deaminate adenine base nearby, which results in
formation of hypoxanthine bases as a part of newly formed inosine
nucleosides instead adenosines, respectively”. There should be told
about numerous mutation combinations, ADARs and other enzymes
including those acting on DNA and/or DNA/RNA-dependent enzymes. Where does this change occur? In which tissues? In which environmental changes or conditions? In which cellular compartment is necessary to debate to discuss further
“RNA editing before splicing”. Also, it is important to mention
whether RNA editing load is higher on intron genes compared to
intronless genes. Xuan et al. (2014) show molecular evidence of RNA
editing in Bombyx chemosensory protein (CSP) genes. Higher load of
mutations is found for intronless genes (see Xuan et al., 2014; 9(2):
e86932). I guess this is not the unique feature and the point and other
studies need to be mentioned when mentioning about RNA editing before
splicing. See also Picimbon (2014) RNA mutations: source of life,
compares editing and splicing.
https://www.walshmedicalmedia.com/abstract/rna-mutations-source-of-life-9533.html

“Literally all RNAs are subject to adenosine deamination of sites
suitable for ADARs in the nucleus” (line 40) is rather dubious.

To my understanding, ADARs recognize specific structures or duplexes. ADARs recognize short, imperfect RNA duplexes and deaminate select
adenosines (specific RNA editing). In contrast, promiscuous editing of
multiple adenosines, also known as 'hyper-editing', usually occurs in
long, perfectly paired RNA duplexes.

The review should be more concise, more clear, better written.

The segregation between protostomes and deuterostomes re. zoology is rather confusing here. How does this relate to genetics, diversity of intron
genes and splicing mechanisms? There are more interesting to say about
insect genetics than just the fact that they are considered as
protostomes, unless more information about development is provided. Which ADAR is found in insects? Is it a rather conserved version of ADAR or an evolutionary mark of insects? Even if the evolution of the ADAR protein family was recently reviewed and is not in the scope of this paper (line 59), this information should be mentioned because the authors should show a clear correlation between ADAR diversity (and thereby diversity of RNA editing) and patterns of splicing dynamics.

A strong criticism here is the lack of clarity about the description
of editing and splicing mechanisms, the locations of such mechanisms
are rather unclear, we jump from the nucleus to cytosol and back.

This should be made clear before to jump from editing to splicing, and
back.

For review, it takes more than one reference to address the functional
diversity (of genes?) reached by alternative splicing and
self-editing. Please make efforts to describe the literature properly,
and be more concise in your remarks and sentences. The style should be
thoroughly revised. The sentence “We began this story with a description of the ubiquitary (?) nature of RNA editing by ADARs”
(line 88) is rather inappropriate for a scientific journal.

Not only the proteome, but also the transcriptomic data should be mentioned to debate about the number of true mutation per tissue or organisms. True or fake, RNA mutations are an eternal debate, however, the authors cannot avoid the debate here. Citations, including Xuan et al., 2014,
2017, 2019, should be part of discussion as they report much higher
diversity than previously described (see also StLaurent et al., 2013
in fly brain; Liu et al., 2020 in bee brain). The citations are
important as mutations or substitutions change gating property and
conductivity of ion channels (see line 101-incomplete). The impact of
mutations on tissue development is matter of introduction towards
genetic diseases and cancers. RNA editing to tissue development and cancer; the gap is missing in this early version.

There are also too many abbreviations that are used without
explanation and make text rather tedious.

Two decades of genomic and post genomic studies of A-to-I RNA editing
…….. what does it mean? Epigenetic events? How does the epigenetic
changes or regulations help recode the protein structure?
Is it the case since the authors reflect only on RNA editing before
splicing? Are the two processes combined in the same tissue or
developmental stage? These questions need to be answered before to
tackle RNA editing (and splicing) for cancer. Figure 1 is rather
confusing since we never know what ADAR (ADAR1) is made of, where it
is expressed and what duplexes it might regulate in humans. ADAR
(ADAR1) should be figure 1. The present figure 1 (negative feedback
loop) is rather hard to understand, there are no numbers mentioned
that should help us to follow the steps. Where does it start? Step1?
If the authors aim to debate about cancer, why do they include viral
infection in the negative loop? Normal conditions versus disease
(which cancer? Which tissue? Which gene affected?) conditions should
be significant matter of text and figure. Interdependence of A-to-I
editing and splicing of mRNA should also be significant matter of
figure to gain in clarity and focus. The present figure 2 is rather
unsuitable for publication. The whole part about synergy, ADAR
(ADAR1), splicing and cancer therapy should be reconsidered.

Reviewer 3 Report

Manuscript ID: ijms-1673373

Interplay between A-to-I editing and splicing of RNA: a potential point of application for cancer therapy

Anton O. Goncharov, Victoria O. Shender, Ksenia G. Kuznetsova, Anna A. Kliuchnikova and Sergei A. Moshkovskii

The authors describe interplay between editing and splicing of RNA in this review article. Recently it has been well accepted that RNA splicing is connected and affected with other gene expression steps. To this end, this review article is interesting and beneficial to many readers of IJMD. However, I have some concerns and suggestions described below.

1) The authors claim that editing precedes splicing machinery on pre-mRNAs. Does this mean editing takes place even with the latter introns before the first intron is removed? As far as I know, I do not know such papers. Please add references in the text.

2) it would be nicer if the authors could add the Figure that shows ADAR proteins schematically with their domain structure information.

3) Although the authors describe many splicing regulatory chemicals, I do not understand the link between them and editing by ADARs. It is required to show ADAR inhibitors, if any, and their effects on both editing and splicing, or the authors should include the effects of those compounds on editing. Otherwise, I strongly recommend the rearrangement of the manuscript with the contents.

Round 2

Reviewer 2 Report

The authors should consider a section of Molecular Mechanisms and Therapies of Colorectal Cancer, Novel Approaches of Anticancer Therapies, Advances in Molecular Genetics of Brain Tumors rather than Multiomics Approaches in Biomedicine to best suit with IJMS (not IJMD).

I still think that the authors should write abstract more cautiously. Line 12: "Adenosine-to-inosine RNA editing is an ubiquitous post-transcriptional modification which is catalyzed by ADAR enzymes and occurs mostly in double-stranded RNA (dsRNA) before splicing." RNA editing is ubiquitous among true metazoans, but not among all tissues. See inosinome atlas (2015). So the authors should be more precise when using ubiquitous for RNA editing. "Dysregulated RNA splicing in cancer often leads to release of intronic RNA to the cytosol, where excessive levels of dsRNA eliciting antiviral-like responses, such as type I interferon signaling" see lines 18-19 (please rewrite). There should be some words on ADAR1 in abstract before to jump on this enzyme for the main propose (see lines 23-27), or remove ADAR1 and only use ADAR.

In text, there should be some more information on ADAR structure, expression and function, in particular on ADAR1 structure, expression (not ubiquitous) and function. Line 66: "Structurally, the enzyme contains one or more dsRNA binding domains, a catalytic deaminase domain and, optionally, Z-DNA binding domain with a function unknown until recently (Figure 1)[8]." Is Z-DNA binding domain just an option for ADAR1? It should be mentioned somewhere that insect adar is different than mammalian ADARs (see Figure 1, lines 110-111). It should also be mentioned somewhere why ADAR2 gains attention on Figure 3 (physical competition?), while the entire text focuses or aims to focus on ADAR1. In this prospect, ADAR1 vs ADAR2, the text (and figure) should gain more clarity. Figure 3: ADAR2, Figure 4: ADAR1.

The new figure 1 is a very thoughtful and very relevant complement for ADAR introduction. Title and legend should be written accordingly (presently there is no title for figure 1). In general, the legend (and title) should be written more cautiously throughout the whole manuscript (remove A; remove The...). Figure 2 now reads well. Please make sure to remove the older version of Figure2 in text (line 207). Same remark for Figure 4, one version of the figure (line 453) should be removed.

"Evolution of the ADAR protein family itself as well as an evolutionary role of ADAR editing were recently reviewed in many papers [2,9,28,29] and generally are not in the scope of this paper focused of the role of RNA processing pathways in mammalian 
cancer." see lines 118-121/ DELETE

The last remark addresses the last sentence (lines 464-467):

"In support of our suggestions, an experimental proof of the hypothesis was published during preparation of this paper. Co-inhibition of ADAR1 and hnRNPC, a recognized splicing regulator, by gene knockouts dramatically increased type I interferon induction, which was beneficial for potential cancer immunotherapy [108]."

Please rewrite to "In support of our suggestions, an experimental proof of our hypothesis (which is.....) was recently brought: co-inhibition of ADAR1 and hnRNPC splicing factor following gene knockout dramatically increased type I interferon induction. It would be particularly beneficial to design new potential cancer immunotherapy [108, and this paper]."

was published during preparation of this paper?

reference dates back to 2021, please rewrite in state-of-art.

Author Response

We thank the Reviewer for fast and useful comments to our paper which obviously helped to improve it significantly. We have agreed to all suggestions of this review round. Below we provide specific replies to the comments. 

Point 1. I still think that the authors should write abstract more cautiously. Line 12: "Adenosine-to-inosine RNA editing is an ubiquitous post-transcriptional modification which is catalyzed by ADAR enzymes and occurs mostly in double-stranded RNA (dsRNA) before splicing." RNA editing is ubiquitous among true metazoans, but not among all tissues. See inosinome atlas (2015). So the authors should be more precise when using ubiquitous for RNA editing. "Dysregulated RNA splicing in cancer often leads to release of intronic RNA to the cytosol, where excessive levels of dsRNA eliciting antiviral-like responses, such as type I interferon signaling" see lines 18-19 (please rewrite). There should be some words on ADAR1 in abstract before to jump on this enzyme for the main propose (see lines 23-27), or remove ADAR1 and only use ADAR.

Response 1. 

Agree. The abstract has been changed. Specific indication on ADAR1 has been removed. 

Point 2. In text, there should be some more information on ADAR structure, expression and function, in particular on ADAR1 structure, expression (not ubiquitous) and function. Line 66: "Structurally, the enzyme contains one or more dsRNA binding domains, a catalytic deaminase domain and, optionally, Z-DNA binding domain with a function unknown until recently (Figure 1)[8]." Is Z-DNA binding domain just an option for ADAR1? It should be mentioned somewhere that insect adar is different than mammalian ADARs (see Figure 1, lines 110-111). It should also be mentioned somewhere why ADAR2 gains attention on Figure 3 (physical competition?), while the entire text focuses or aims to focus on ADAR1. In this prospect, ADAR1 vs ADAR2, the text (and figure) should gain more clarity. Figure 3: ADAR2, Figure 4: ADAR1.

Response 2.

Agree. A phrase was added explaining a presence of Z-DNA domain in ADAR1 isoforms, with reference to explanatory Fig.1 (lines 94-96). Below, in Section 2, a newly discovered role of Z-alpha domain of p150 ADAR1 isoform was described (lines 227-230 - hereinafter for a version with tracked changes). More explanations of Fig.3 were added to the legend. There, ADAR means both isoforms, ADAR2 reflects a specific fact of its competition to the U2 spliceosome component [73] (lines 435-437). 

Point 3. The new figure 1 is a very thoughtful and very relevant complement for ADAR introduction. Title and legend should be written accordingly (presently there is no title for figure 1). In general, the legend (and title) should be written more cautiously throughout the whole manuscript (remove A; remove The...). Figure 2 now reads well. Please make sure to remove the older version of Figure2 in text (line 207). Same remark for Figure 4, one version of the figure (line 453) should be removed.

Response 3. 

Agree. The legend was missed by a technical mistake and now corrected. Legends to other figures are also checked and modified. Further, in the version with accepted changes, the extra figures disappeared. See please both versions of the paper, with tracked and accepted changes.  

Point 4. "Evolution of the ADAR protein family itself as well as an evolutionary role of ADAR editing were recently reviewed in many papers [2,9,28,29] and generally are not in the scope of this paper focused of the role of RNA processing pathways in mammalian 

cancer." see lines 118-121/ DELETE

Response 4.

Agree. This paragraph was deleted. On the other hand, we do not want to miss the references which may be useful for further reading. Thus, we inserted a short phrase (lines 113-114).  

Point 5. The last remark addresses the last sentence (lines 464-467):

"In support of our suggestions, an experimental proof of the hypothesis was published during preparation of this paper. Co-inhibition of ADAR1 and hnRNPC, a recognized splicing regulator, by gene knockouts dramatically increased type I interferon induction, which was beneficial for potential cancer immunotherapy [108]."

Please rewrite to "In support of our suggestions, an experimental proof of our hypothesis (which is.....) was recently brought: co-inhibition of ADAR1 and hnRNPC splicing factor following gene knockout dramatically increased type I interferon induction. It would be particularly beneficial to design new potential cancer immunotherapy [108, and this paper]."

was published during preparation of this paper?

reference dates back to 2021, please rewrite in state-of-art.

Response 5.

Agree. The reference and its description was added to Section 2 (lines 265-269). The final paragraph was rewritten according to the Reviewer's comment. 

Reviewer 3 Report

The authors addressed to all of my concerns and suggestions successfully. I do not have any more comments and suggestions before accept.

Author Response

We thank the Reviewer for the consideration of our work.